# Effect of Vaccination against *Glässer*’s Disease in a Farm Suffering from Polyserositis in Weaned Pigs

**DOI:** 10.3390/vetsci9120691

**Published:** 2022-12-12

**Authors:** Jasmine Hattab, Giuseppe Marruchella, Abigail Rose Trachtman, Luigino Gabrielli, Nicola Bernabò, Francesco Mosca, Pietro Giorgio Tiscar

**Affiliations:** 1Department of Veterinary Medicine, University of Teramo, Loc. Piano d’Accio, 64100 Teramo, Italy; 2Veterinary Practitioner, 63073 Offida, Italy; 3Department of Bioscience and Agro-Food and Environmental Technology, University of Teramo, via Renato Balzarini 1, 64100 Teramo, Italy

**Keywords:** swine, polyserositis, etiology, Mycoplasma hyorhinis, Glaesserella parasuis, Streptococcus suis, vaccine

## Abstract

**Simple Summary:**

This study mainly aims to investigate the impact and the etiology of swine polyserositis, focusing on three causative agents: *Glaesserella parasuis*, *Streptococcus suis* and *Mycoplasma hyorhinis*. Results indicate that the etiology of polyserositis is an intricate puzzle of pathogens, and that each pig herd likely represents a unique scenario. Therefore, the correct diagnosis of polyserositis can be really challenging and often makes the implementation of therapeutic and control strategies frustrating.

**Abstract:**

Polyserositis mostly affects 4–8 weeks old piglets and is usually caused by *Glaesserella parasuis,* and/or *Streptococcus suis,* and/or *Mycoplasma hyorhinis.* The present study aimed to investigate the prevalence and etiology of polyserositis in a tricky pig herd. The concurrent effect of vaccination for *Glässer*’s disease was also assessed. A total of 46 sows and 387 piglets were herein investigated, subdivided into three groups based on their immune status (i.e., vaccination of sows and piglets). All the piglets found spontaneously dead between the 2nd and 16th week of age were recorded and necropsied. Whenever polyserositis was diagnosed, biomolecular investigations were carried out to detect the above-mentioned pathogens. *Mycoplasma hyorhinis* was detected most frequently (n = 23), often as the only causative agent (n = 15), whereas *S. suis* was observed in 8 cases (6 as the only pathogen). Moreover, *Glaesserella parasuis* was demonstrated in 6 piglets, always in combination with *Mycoplasma hyorhinis* and/or *Streptococcus suis*. Vaccination did not significantly affect mortality rates. Overall, our data indicate that polyserositis is likely caused by an intricate puzzle of pathogens, even when dealing with a small herd and during a short time span. That makes it challenging to achieve the correct diagnosis and to properly manage this health issue.

## 1. Introduction

Serous membranes (i.e., the pericardium, pleura, and peritoneum) are prone to inflammatory injuries, which are characterized by the effusion of fibrinous and/or suppurative exudates. Serositis is usually due to the hematogenous spreading of bacterial pathogens. Less commonly, it can result from penetrating wounds or direct extension of adjacent inflammatory foci [1].

The concurrent inflammation of two or more serous membranes is called polyserositis and represents a widespread and economically relevant disease condition in modern pig farming. Swine polyserositis mostly affects 4–8 weeks old piglets and has long been identified with *Glässer*’s disease by *Glaesserella* (previously known as *“Haemophilus”*) *parasuis* (*G. parasuis*) [2]. However, *Streptococcus suis* (*S. suis*) [3] and *Mycoplasma hyorhinis* (*M. hyorhinis*) [4] can often induce similar disease conditions. To a lesser extent, polyserositis can be caused by other pathogens, such as *Escherichia coli* [5], *Pasteurella multocida* [6], and *Actinobacillus pleuropneumoniae* [7]. The clinical onset of polyserositis is usually sudden and often includes neurological signs and swollen leg joints, due to the concurrent localization of the pathogen in the central nervous system and synovial membranes [8].

Recently, Salogni et al. [9] carried out a large-scale study (154 pigs from 80 herds) about the etiology of swine polyserositis in Northern Italy, showing that *G. parasuis* and *M. hyorhinis* were most frequently detected at the same prevalence rate. In our opinion, the paper by Salogni et al. provides a useful and reliable picture of a high-density breeding area, although unlikely representative of each pig herd. Considering that, the present study aimed to investigate the prevalence and etiology of polyserositis in a tricky pig herd under strict field conditions. The concurrent effect of vaccination for *Glässer*’s disease was also assessed.

## 2. Materials and Methods

The study was carried out in a farrow-to-finish pig herd located in Central Italy. The breeding stock consisted of 70 sows (Landrace x Large White x Duroc) and 3 boars (Mora Romagnola breed). Piglets were weaned at 4 weeks and reared indoors until 16 weeks of age. Thereafter, pigs were moved outdoors until they reached the market weight of 160 kgs.

Sows were vaccinated against colibacillosis and clostridiosis (Suiseng, Hipra, Girona, Spain), whereas three weeks old piglets were vaccinated against porcine circovirus type 2 (PCV-2), *Mycoplasma hyopneumoniae*, and porcine reproductive and respiratory syndrome virus (PRRSv; 3FLEX^®®^, Boehringer Ingelheim, Ingelheim am Rhein, Germany) according to the manufacturer’s recommendations. Compulsory vaccination against *Aujeszky*’s disease (Aujeszky A-Suivax GI, Fatro, Bologna, Italy) was implemented as per Italian law.

Newly weaned piglets were fed a diet containing amoxicillin for one week (50 g/100 kgs of food), whereas sick piglets were usually treated with amoxicillin by intramuscular injection (Longocillina L.A. 150, 1 mL/10 kgs on alternate days; Ceva Salute Animale S.p.A., Agrate Brianza, Italy).

During the few months prior to the study, postweaning mortality rates were very high and several cases of *Glässer*’s disease and streptococcosis had been diagnosed, as confirmed by culture of *G. parasuis* and *S. suis*, respectively (data provided to the farmer by the “*Istituto Zooprofilattico Sperimentale dell’Umbria e delle Marche*” in the framework of its routine diagnostic activities).

Between September 2020 and March 2021, a total of 46 sows and 387 piglets were included in the present investigation. During this period, the Veterinary Practitioner decided to vaccinate sows and weaned piglets for *Glässer*’s disease (Porsilis Glasser, MSD Animal Health, Rahway, NJ, USA; see Table 1 for details). Sows were vaccinated twice, 6–8 and 4 weeks before farrowing, whereas piglets were vaccinated only once at 4–5 weeks of age.

As a routine, piglets from the same litter were identified by means of ear tags. All piglets spontaneously dead between the 2nd and 16th week of age were recorded and As a routine, piglets from the same litter were identified by means of ear tags. All piglets spontaneously dead between the 2nd and 16th week of age were recorded and necropsied on a weekly basis, upon freezing whenever needed. Samples (i.e., pericardium, heart, and fibrin) were taken from all polyserositis-affected pigs, placed into single tubes, kept on ice, and delivered to the laboratory (Veterinary Teaching Hospital, University of Teramo, Italy).

Total DNA was extracted using ExgeneTM Tissue SV (GeneAll^®^, Seoul, South Korea) according to the manufacturer’s protocol. Biomolecular investigations (i.e., polymerase chain reactions, PCR) were carried out to detect the most common causative agents of polyserositis—namely, *G*. *parasuis*, *M*. *hyorhinis*, *S*. *suis*—following previously published protocols [10,11,12,13]. PCR reactions were prepared as a total volume of 25 µL, consisting of 4 µL of template DNA, 12.5 µL of master mix (GoTaq^®^ Green Master Mix, Promega Corporation, Madison, WI, USA), and 0.5 µL of forward and 0.5 µL of reverse primers (see Table 2 for details).

Positive and negative controls were included in each PCR run. All *G. parasuis*-positive samples were further tested for serotyping by means of previously described PCR protocols [14,15].

Finally, data were submitted to statistical analysis. Fisher’s exact test was used to compare the mortality rates, as well as the prevalence of different pathogens among groups. The following cut-off points were established for significance: values differ significantly at *p* ≤ 0.05 and at the trend level at 0.05 < *p* < 0.1.

## 3. Results

The weekly rates of total mortality are shown in Figure 1.

During the study period, the overall mortality rate was 10.6% in suckling piglets (41 out of 387 piglets), most deaths being recorded during the first week of age (35 out of 41 dead piglets). After weaning, the overall mortality rate was 19.9% (69 out of 346 weaned piglets), with a prominent peak at 8 weeks of age (see Table 3 for details).

At necropsy, polyserositis was frequently diagnosed in weaned piglets (39 out of 69 cadavers), whereas it was never observed in suckling piglets (Figure 2).

As shown in Figure 3, most cases of polyserositis affected 6-to-8- and 15-week-old piglets.

No significant difference was detected among the three groups regarding polyserositis-specific (i.e., mortality due to polyserositis) and polyserositis-proportionate (i.e., the proportion of deaths attributable to polyserositis) mortality rates (see Table 3 for further details).

The results of the PCR tests are graphically shown in Figure 4. Overall, the genome of *M*. *hyorhinis*, *S*. *suis* and *G*. *parasuis* was detected in 23, 8, and 6 polyserositis-affected piglets, respectively. *G*. *parasuis* was always detected in combination with other pathogens. On the contrary, *M*. *hyorhinis* and *S*. *suis* were detected as the only causative agent in 15 and 5 piglets, respectively.

As shown in Figure 5, *M. hyorhinis* and *S. suis* were observed during almost the entire postweaning period, with peaks of incidence at 8 and 15 weeks of age, respectively. *G. parasuis* was detected between 8 and 12 weeks of age, the highest incidence being observed at 8 weeks of age.

Overall, the detection of *M. hyorhinis* was significantly higher when compared with *G. parasuis* (*p* < 0.01) and *S. suis* (*p* < 0.01). In particular, the detection of *M. hyorhinis* was significantly more frequent than *G. parasuis* in group B (*p* = 0.02) and group C (*p* = 0.01).

A single piglet from group B proved to be positive for *G. parasuis*, whereas *G. parasuis* was never detected in group C. The detection of *G. parasuis* was lower in group C at trend level (*p* = 0.05) when compared with group A. Moreover, the occurrence of co-infections was significantly lower in group C when compared with group A (*p* = 0.02).

Biomolecular tests for serotyping *G. parasuis* always yielded negative results.

## 4. Discussion

Although mortality is among the foremost causes of economic loss in livestock production, it is difficult to estimate its physiologic rate in pig farming. Preweaning mortality usually ranges between 10% and 20% and is affected by a vast number of interrelated factors (e.g., birth weight, litter size, heating of farrowing crates, sow health, etc.) [16,17]. In the USA, a mortality rate of 3.6% was reported in nursery pigs after a long large-scale data collection [18]. Considering that, the pig herd under study showed an acceptable rate of preweaning mortality, which was mostly restricted to the first week of age, as commonly observed worldwide [17]. Conversely, postweaning mortality was undoubtedly too high, with most of the deaths falling in the critical period between 7-to-9 weeks of age, when passive immunity decreases, and the development of self-antibodies is not yet effective [19,20].

The present study confirms that polyserositis can frequently occur in weaned piglets, and it is likely caused by an intricate puzzle of pathogens, even when dealing with a small herd and during a brief time span.

Over recent years, rising relevance has been given to *M. hyorhinis*, at least partially due to the widespread use of sensitive diagnostic tools, which overcome the difficulty of culturing this pathogen in vitro. *M. hyorhinis* is a commensal bacterium in swine and mainly causes polyserositis and polyarthritis in nursery pigs between 8 and 10 weeks of age [21]. Based on our data, *M. hyorhinis* could play a major role in the etiology of polyserositis, alone or in combination with other pathogens, and during a longer time span. Reasonably, the administration of beta-lactams held off the infections by *S. suis* and *G. parasuis* and further increased the proportionate prevalence of *M. hyorhinis*-induced polyserositis. Although vaccination against *M. hyorhinis* can reduce clinical signs and lesions [22], commercial vaccines are currently unavailable in Europe. Therefore, *M. hyorhinis*-induced diseases are usually treated with antimicrobials [23]. A recent study carried out in European countries (including Italy) indicates that doxycycline, oxytetracycline, and tiamulin are the most effective compounds, at least in vitro [24].

Vaccination can reduce mortality by *Glässer*’s disease. The efficacy of *G. parasuis* vaccines is serovar-specific, cross-protection being debated and likely limited [25,26,27]. In Italy, *G. parasuis* serovar 4 was reported as the most prevalent one, followed by serovar 13 and serovar 5, whereas about 27% of strains were non-typeable through agar gel diffusion test [28]. In the present study, *G. parasuis* serovars remained unidentified, and vaccination weakly affected mortality rates. Reasonably, the presence of *M. hyorhinis* as the main causative agent accounted for the low efficacy of vaccination, whereas *G. parasuis* played a less relevant role. However, we remark that *G. parasuis* was never detected in vaccinated piglets, and vaccination significantly reduced the occurrence of co-infections.

Herein, a high portion of cases of polyserositis (about 28%) proved to be negative for *M. hyorhinis*, *G. parasuis*, and *S. suis* and were likely caused by other pathogens, which should be further investigated. Unfortunately, additional bacteriological investigations were seldom carried out, and *Trueperella pyogenes* was cultured in a single piglet (data not shown). Moreover, the causative role of viruses (e.g., PRRSv, PCV-2) should be considered, as they often predispose to bacterial infections. In particular, PRRSv often contributes to the etiology of polyserositis, whereas the impact of PCV-2 is likely mitigated by the widespread use of effective vaccines [9,29].

The multi-faceted etiology of polyserositis greatly complicates its control and therapy. It seems that targeting a single pathogen (by means of vaccine and/or antimicrobials) might change the balance among causative agents without affecting the severity of polyserositis. As an example, herein the administration of amoxicillin and the vaccination for *Glässer*’s disease likely limited the role of *S. suis* and *G. parasuis* while enhancing the etiological relevance of *M. hyorhinis*. Therefore, the effective control of polyserositis should always include the improvement of herd management, minimizing stressful situations that worsen most diseases in intensive pig farming [8].

## 5. Conclusions

The present study confirms that the etiology of polyserositis is multi-factorial and suggests that each pig herd represents a unique scenario. A suitable diagnostic approach is crucial to properly manage this health issue, aiming to counteract the most relevant pathogens. However, our data indicate that this could be challenging, as “one-shot” investigation on a single or few animals might be insufficient, if not misleading. In our opinion, *M. hyorhinis* should be regarded as an “emerging” and relevant pathogen. Vaccination against *Glässer*’s disease could be effective when *G. parasuis* plays a key etiological role.

## Figures and Tables

**Figure 1 vetsci-09-00691-f001:**
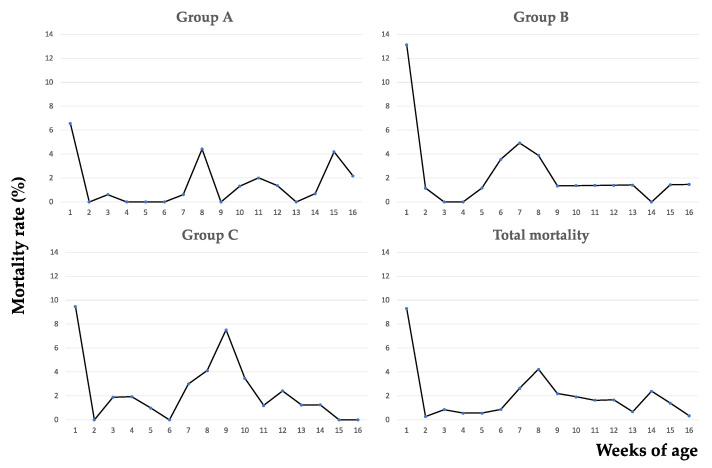
Weekly-based mortality rates. All groups show a peak of mortality around 8 weeks of age. In group A, an additional peak is observed at 15 weeks of age.

**Figure 2 vetsci-09-00691-f002:**
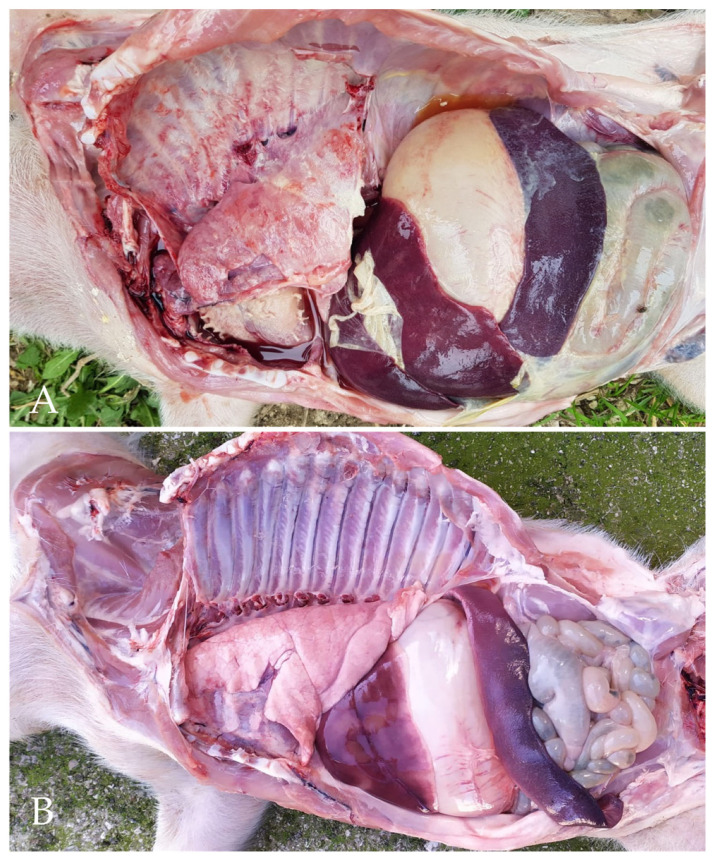
Piglets. Necropsy. (**A**) Polyserositis-affected piglet. Large amounts of fibrin are seen upon both pleural sheets, the epicardium, as well as the peritoneum lining the spleen and the liver. (**B**) A piglet dead with no evidence of polyserositis is shown for comparison.

**Figure 3 vetsci-09-00691-f003:**
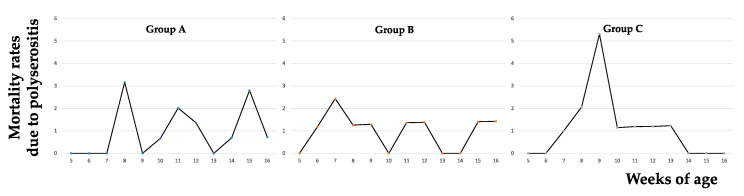
Postweaning mortality due to polyserositis. The trends of polyserositis-specific mortality largely overlap those shown in Figure 1. In group (**A**), and to a lesser extent in group (**B**), mortality shows a swinging pattern, whereas it is mostly restricted to at 8-9 weeks of age in group (**C**).

**Figure 4 vetsci-09-00691-f004:**
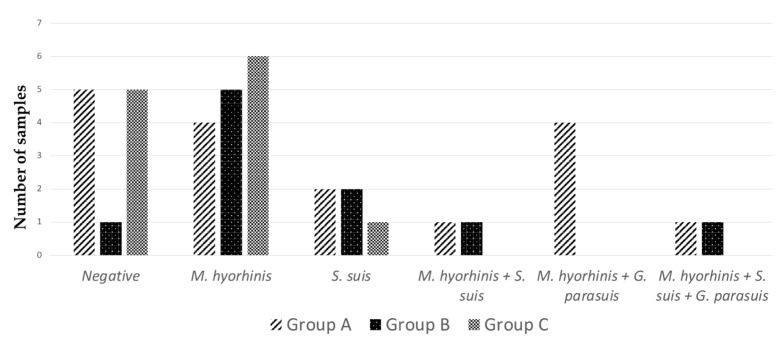
Results of biomolecular investigations. *M. hyorhinis* was detected most frequently, alone or in combination with other pathogens. Conversely, *G. parasuis* was seldom detected. In 11 cases, PCR tests yielded negative results.

**Figure 5 vetsci-09-00691-f005:**
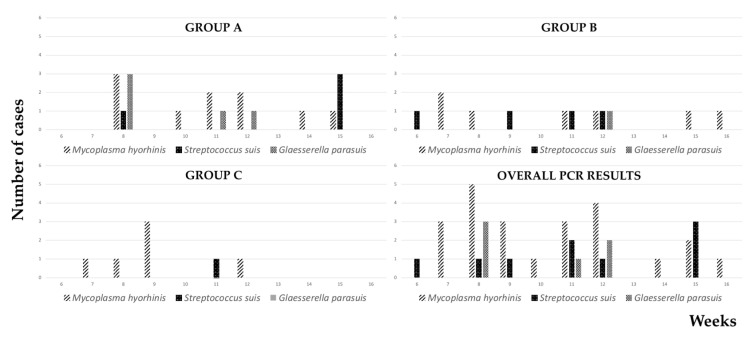
Weekly-based results of biomolecular investigations. Overall, *M. hyorhinis* was detected from the 7th to the 16th week of age, with the only exception of the 13th week. *S. suis* infection was unevenly demonstrated, whereas *G. parasuis* was mostly observed at 8 weeks of age in group A.

**Table 1 vetsci-09-00691-t001:** **Animals included in the present study.** Based on vaccination for *Glasser*’s disease, three groups were distinguished: (**A**) non-vaccinated piglets born to non-vaccinated sows; (**B**) non-vaccinated piglets born from vaccinated sows; and (**C**) vaccinated piglets born to vaccinated sows.

Group	Sows (Number)		Piglets (Number)	
	Vaccinated	Not vaccinated	Vaccinated	Not Vaccinated
A	//	19	//	172
B	12	//	//	99
C	15	//	116	//

**Table 2 vetsci-09-00691-t002:** Primers used for PCR tests.

Primer Name	Primer Sequence (5′-3′)	Amplicon Size (bp)	Targeted Pathogen	Reference
Hps-f	GTGATGAGGAAGGGTGRTGT	822	*Glaesserella parasuis*	[10]
Hps-r	GGCTTCGTCRCCCTCTGT			
YADAF 1	TTTAGGTAAAGATAAGCAAGGAAATCC	406	*Glaesserella parasuis;* vtA domain group 1	[11]
PADHR 1	CCACACAAAACCTACCCCTCCTCC			
YADAF 2	AGCTTAATATCTCAGCACAAGGTGC	294	*Glaesserella parasuis;* vtA domain group 2	[11]
PADHR 1	CCACACAAAACCTACCCCTCCTCC			
YADAF 3	AATGGTAGCCAGTTGTATAATGTTGC	291	*Glaesserella parasuis;* vtA domain group 3	[11]
PADHR 1	CCACACAAAACCTACCCCTCCTCC			
Mhr_p37 f	TTCTATTTTCATCTATATTTTCGC	101	*Mycoplasma hyorhinis*	[12]
Mhr_p37 r	TCATTGACCTTGACTAACTG			
JP4	GCAGCGTATTCTGTCAAACG	688	*Streptococcus suis*	[13]
JP5	CCATGGACAGATAAAGATGG			

**Table 3 vetsci-09-00691-t003:** Postweaning mortality rates. Total mortality rate was higher in group C (*) at a trend level (*p* = 0.078) when compared with group A.

	Group A	Group B	Group C
Total mortality rate	15.72	22.09	24.75 *
Polyserositis-specific mortality rate	10.69	11.62	11.88
Polyserositis-proportionate mortality	68.00	52.63	48.00

## Data Availability

Not applicable.

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
