# Peer review of "Effect of Vaccination against Glässer’s Disease in a Farm Suffering from Polyserositis in Weaned Pigs"

_vetsci, 2022, doi:10.3390/vetsci9120691_

Round 1

Reviewer 1 Report

This study mainly aims to investigate the impact and the etiology of swine poly serositis. Results indicate that the etiology of polyserositis is an intricate puzzle of pathogens. Although the topic is interesting, big concerns need to be addressed before publication.

1. Some bacterial name which was first mentioned should be used the full name, not the short name. The authors need to carefully checked in the manuscript.

2. Line 52 -59. The authors should compared and discussed the difference and similarity between this study and the reference 9 content.

3. Line 66-69. How and when the PCV-2, Mycoplasma hyopneumoniae and PRRSV were vaccinated.

4. Line 76 to 78. The S. suis and the G. parasuis were the commen upper respiratory tract bacterial. The authors in this study diagnosed the diseases by culture of G. parasuis and S. suis. How the authors distinguish these two bactrials. Also whether the authors were tested the serum from the pigs?

5.Figure 1 and Figure 3.  Has the author detected the difference? Using the Logrank test?

6. Figure 2. The author should add the control group.

7. Figure 4.  What is the mean of the Y-axis?

8. Figure 5. What are the differences of Polyserositis infected by those bacterial.

The author should be deeply discussed.

9. References. The author should be cited the Standard references style.

Reviewer 2 Report

The manuscript is generally well written, but the English does require some attention.

I think it is a useful contribution, because although the sample of pigs was small, the results were informative, and it is quite logical that when certain common pathogens are controlled, a commensal but potentially pathogenic agent should emerge during times when piglets are particularly vulnerable. Certainly the owner should be looking at management aspects that may be causing stress in the affected age groups.

I have not come across the expression “tricky pig herd”. I would expect it to mean that the pig herd was in some way challenging in terms of its health status or performance, but there is no explanation of why the herd is described as “tricky”. The authors should provide an explanation, as they may be using the word incorrectly.

Edits required:

Lines 29, 227: data = plural form of datum, so verb should be indicate

Line 40: should read ‘it can result…’, as serositis is singular

Line 203: replace ‘warranted’ with ‘accounted for’ – warranted means called for justified, e.g. more investigation is warranted, but it is being used incorrectly in this context.

Author Response

Figure 2 has been changed, as suggested by the Reviewer.

Round 2

Reviewer 1 Report

I still think that the authors should add the not observerd  polyserositis figure, so that the readers could understand the differences.

Author Response

None